# Development of a Multiplex RT-PCR Assay for Simultaneous Detection of Four Potential Zoonotic Swine RNA Viruses

**DOI:** 10.3390/vetsci9040176

**Published:** 2022-04-07

**Authors:** Gebremeskel Mamu Werid, He Zhang, Yassein M. Ibrahim, Yu Pan, Lin Zhang, Yunfei Xu, Wenli Zhang, Wei Wang, Hongyan Chen, Lizhi Fu, Yue Wang

**Affiliations:** 1Heilongjiang Provincial Key Laboratory of Animal and Comparative Medicine, State Key Laboratory of Veterinary Biotechnology, Harbin Veterinary Research Institute, Chinese Academy of Agricultural Sciences, Harbin 150069, China; ashenafymamo@gmail.com (G.M.W.); zhanghe3789@163.com (H.Z.); yassin8322@gmail.com (Y.M.I.); gudaoqiusheng37@163.com (Y.P.); zhanglin19920402@163.com (L.Z.); xuyunfei202203@163.com (Y.X.); zwl5561@163.com (W.Z.); wangwei02@caas.cn (W.W.); chenhongyan@caas.cn (H.C.); 2Chongqing Academy of Animal Science, Chongqing 408599, China; 3College of Veterinary Medicine, Southwest University, Chongqing 400715, China

**Keywords:** multiplex PCR, Swine potential zoonotic viruses, sapovirus, encephalomyocarditis virus, rotavirus A and astrovirus

## Abstract

Swine viruses like porcine sapovirus (SaV), porcine encephalomyocarditis virus (EMCV), porcine rotavirus A (RVA) and porcine astroviruses (AstV) are potentially zoonotic viruses or suspected of potential zoonosis. These viruses have been detected in pigs with or without clinical signs and often occur as coinfections. Despite the potential public health risks, no assay for detecting them all at once has been developed. Hence, in this study, a multiplex RT-PCR (mRT-PCR) assay was developed for the simultaneous detection of SaV, EMCV, RVA and AstV from swine fecal samples. The PCR parameters were optimized using specific primers for each target virus. The assay’s sensitivity, specificity, reproducibility, and application to field samples have been evaluated. Using a pool of plasmids containing the respective viral target fragments as a template, the developed mRT-PCR successfully detected 2.5 × 10^3^ copies of each target virus. The assay’s specificity was tested using six other swine viruses as a template and did not show any cross-reactivity. A total of 280 field samples were tested with the developed mRT-PCR assay. Positive rates for SaV, EMCV, RVA, and AstV were found to be 24.6% (69/280), 5% (14/280), 4.3% (12/280), and 17.5% (49/280), respectively. Compared to performing separate assays for each virus, this mRT-PCR assay is a simple, rapid, and cost-effective method for detecting mixed or single infections of SaV, EMCV, RVA, and AstV.

## 1. Introduction 

Swine viruses cause a major economic loss in the swine industry [1,2,3,4,5]. Thus far, more than sixteen viruses have been detected from porcine fecal specimens [6,7,8,9]. From these swine viruses, due to the asymptomatic nature of infections, unless there is a confounding factor or a mixed infection of another pathogenic virus, infections from porcine sapovirus (SaV), encephalomyocarditis virus (EMCV), porcine rotavirus A (RVA) and porcine astrovirus (AstV) might not be noticed [10,11,12,13]. Though the economic impact of SaV, EMCV, RVA and AstV viruses on the swine industry might not be significant enough to attract the attention of pig producers, their potential to become zoonotic diseases warrants proper attention [12]. More importantly, even asymptomatic pigs can shed the virus into the environment or directly transmit it to people at risk [9,10,11,12]. Likewise, SaV, EMCV, RVA, and AstV were found to be genetically closely related to human viruses and hence have a zoonotic potential and/or are suspected of zoonosis [5,11,12,14,15,16,17,18]. 

To develop vaccines against the prevalent viral strains, early and accurate virus detection is needed and hence, early virus detection is an integral part of disease control and prevention programs [19]. However, coinfection complicates early and accurate virus detection. In the presence of coinfection, it is often difficult to know the underlining cause of disease [20,21]. This is because, in the presence of coinfection, viruses that do not usually cause apparent clinical signs during a single infection could contribute to the severity of an existing infection, further complicating disease diagnosis [11,22]. Moreover, compared to other swine viruses, these potential zoonotic viruses, due to their high coinfection rate, higher virus shedding, fecal–oral transmission, asymptomatic nature, and potential to transmit to humans, demands special attention for early detection and accurate identification. 

Virus isolation, electron microscopy, serological tests, and nucleic acid detection methods are the most commonly used methods for detecting viruses [21,23,24]. Recently nucleic acid-based and serological detection methods are being used widely for diagnosing swine viral infections [23,24,25]. Nucleic acid-based detection methods are preferred over serological methods for determining current infection status [26]. The existing nucleic acid-based diagnostic assays for detecting potential zoonotic swine viruses depend on a single PCR. These single PCR-based detection methods consume more time and resources to detect each virus at a one-time point, whereas using multiplex PCR assays, nucleic acid detection methods have another added advantage for the simultaneous detection of viruses [24,25]. Multiplex reverse transcriptase PCR (mRT-PCR) based virus diagnostic assays provide both the affordability and accuracy of virus detection, enabling faster and wider field-based applications [23,26]. In this study, in an attempt to solve challenges associated with detecting potential zoonotic swine viruses, a multiplex PCR assay that could simultaneously detect SaV, EMCV, RVA and AstV from porcine fecal samples has been developed.

## 2. Materials and Methods

### 2.1. Nucleic Acid Extraction and Reverse Transcription

From May to September 2021, 280 fecal samples from pigs of different age groups were collected from the Heilongjiang province in China. Prior to storage, 10% (*w*/*v*) stool suspensions were mixed with PBS (phosphate-buffered saline, 7.4 PH), centrifuged at 400× *g* for 20 min at 4 °C, and collected supernatants. Total RNA was extracted from fecal supernatants using the Tiangen virus RNA extraction kit (Tiangen Biotech, Beijing, China), according to manufacturer instructions. 

Complementary DNA (cDNA) was synthesized in a 20 µL reverse transcription (RT) reaction mixture containing 2 µL of Golden MLV buffer, 1 µL Golden MLV enzyme, 1 µL Random hexamer primers and 1 µL dNTP, 0.5 µL Rnase inhibitor and 1 µg of total RNA and RNase free double distilled water (ddH_2_O) and then incubated at 37 °C for 15 min and at 85 °C for 5 s (TakaRa, Dalian, China). 

### 2.2. Primer Design and Construction of SaV, EMCV, RVA and AstV Plasmids

Nucleotide sequences were downloaded from NCBI and aligned prior to primer design to identify conserved regions for each selected virus. The primers (Table 1) used to amplify the fragments of SaV, EMCV, RVA, and AstV were designed by primer3 [27], and then primer properties such as primer heterodimer, self-dimer and hairpin were checked by OligoAnalyzer (https://www.idtdna.com/calc/analyzer, accessed on 20 June 2020). Each amplicon was purified and cloned into the pMD18-T vector (TaKaRa, Dalian, China). The plasmids were transformed into competent *Escherichia coli* DH5α cells. The plasmids were extracted using a mini plasmid extraction kit (Tiangen, Beijing, China) from a bacterial solution cultured at 37 ℃ for 14–16 h and quantified by a NanoDrop spectrophotometer (Thermo Fisher, Waltham, MA, USA). The plasmid constructs were then confirmed by PCR and sequencing (CometBio, Jilin, China). For mRT-PCR optimization, the plasmids were used as templates.

### 2.3. Single RT-PCR and mRT-PCR Reaction Optimization

Prior to performing mRT-PCR, the designed primers were used to perform single RT-PCR (sRT-PCR) for SaV, EMCV, RVA, and AstV. A 40 µL PCR reaction was prepared using 5 µL of cDNA, 5% DMSO, rTaq enzyme (0.5–1 unit), 0.25 µM of each primer, 1.5 mM MgCl_2_, 0.2 mM dNTP, 4X rTaq buffer and autoclaved ddH_2_O was added to make 40 µL volume. In the negative control reaction, ddH_2_O was used as a template. In a Bio-Rad PCR Thermo Cycler (Bio-Rad, Hercules, CA, USA), the prepared reaction was amplified as follows: one cycle at 94 °C for 5 min; 35 cycles of denaturation at 98 °C for 10 s; gradient annealing (48 °C to 58 °C) for 30 s; 72 °C for 45 s extension; and a final extension step of 10 min at 72 °C. Gel electrophoresis with a 1.5% agarose gel in 1X TAE buffer was used to examine PCR results. 

The mRT-PCR reactions were optimized by varying a single parameter while keeping other parameters constant. Parameter variables including annealing temperature (48 to 58 °C), number of PCR cycle (25 to 40), concentrations of primer (0.05 μM to 0.4 μM), MgCl_2_ (1.0 to 4.0 mM), dNTP (0.3–0.9 mM), and TakaRa rTaq DNA Polymerase (2 to 6 U) were optimized. The PCR products were visualized by electrophoresis through 1.5% agarose gel in 1× TAE buffer. Following mRT-PCR condition optimization, the amplicons were sequenced for confirmation (CometBio, Jilin, China).

### 2.4. Assay Sensitivity, Specificity, and Reproducibility

The sensitivity of sRT-PCR and mRT-PCR was compared using a 10-fold serially diluted standard plasmids of known DNA copy number. The DNA copy number of each standard plasmid was calculated using the formula described in [24]. To serve as a template for the sRT-PCR and mRT-PCR, 10^6^ copies/μL of each cloned virus fragment were mixed into a single tube and diluted 10-fold serially at a concentration gradient of 10^0^ to 10^6^ copies/μL.

Specificity of the developed mRT-PCR was tested by using cDNA of porcine epidemic diarrhea virus (PEDV), Seneca Valley virus (SVV), transmissible gastroenteritis virus (TGEV), porcine respiratory and reproductive syndrome virus (PRRSV), porcine delta coronavirus (PDCoV), porcine sapelovirus (PSV) and porcine enterovirus (PEV) mixed with cDNA of each of the viruses included in the mRT-PCR as templates. The amplicons were purified and sequenced to confirm the specificity of the assay.

To evaluate the intra-assay reproducibility of the developed assay, mRT-PCR reactions were performed in triplicate on each optimized parameter. Furthermore, inter-assay reproducibility was tested by running four different PCR reactions under optimized conditions with freshly prepared templates.

### 2.5. Detecting Target Viruses from Field Samples Using mRT-PCR

The developed mRT-PCR assay was used to test a pool of 280 porcine fecal field samples for SaV, EMCV, RVA, and AstV, and 130 of these samples were also tested using single PCR Primers for each virus.

### 2.6. Phylogenetic Analysis of the Detected Viruses 

Some of the porcine fecal field samples that were mRT-PCR positive for SaV, EMCV, RVA and AstV were reamplified using sRT-PCR. The amplified and purified PCR products were sequenced. MEGA-X program was used to align the nucleotide sequences. The phylogenetic trees were constructed with MEGA-X software using the neighbor-joining method [28], Kimura-2 distances, and a bootstrap sampling technique with 1000 replicates [29].

## 3. Results

### 3.1. Optimized Conditions of the mRT-PCR

Before mRT-PCR reaction optimization, sRT-PCR standardization and optimum annealing temperature determination were done using gradient PCR for each virus at a temperature of 48 to 58 °C. Primers of SaV, EMCV, RVA and AstV produced an amplicon size of 305 bp, 438 bp, 570 bp and 702 bp, respectively. The annealing temperature was optimized at 50–52 °C. After optimizing the annealing temperature, each PCR reaction condition was optimized one at a time by keeping other conditions constant. After repetitive experiments, an optimum concentration of dNTP, MgCl_2_ and Taq polymerase was determined at 0.6 mM, 0.35 mM and 6 units, respectively. Primer concentration were optimized at 0.15, 0.113, 0.15 and 0.1 µM for SaV, EMCV, RVA and AstV, respectively. All reactions were performed at 40 µL reaction volume and 35 cycles. Each of the amplicons was visualized by electrophoresing 10 µL aliquots through 1.5% agarose gels in 1× TAE (40 mM Tris-acetate [pH 8.0], 1 mM EDTA). Using the optimized mRT-PCR conditions, all four viral fragments were amplified successfully (Figure 1A).

### 3.2. Assay Sensitivity

The sensitivity of sRT-PCR and mRT-PCR was tested using ten-fold serial dilutions of the SaV, EMCV, RVA and AstV plasmid constructs. It was found that the developed mRT-PCR assay could simultaneously detect up to 2.5 × 10^3^ copies of each template, while single PCR could detect 2.5 × 10^1^, 2.5 × 10^3^, 2.5 × 10^2^, and 2.5 × 10^2^ copies of SaV, EMCV, RVA and AstV, respectively (Figure 2).

### 3.3. Assay Specificity

The specificity of each primer pair was determined using sRT-PCR and mRT-PCR (Figure 1B). Both sRT-PCR and mRT-PCR were found specific for the target viral agent because no amplicons were produced with other viral agents, including PEDV, SVV, TGEV, PRRSV, PDCoV, and PSV (Figure 1B, lanes 5–10). All positive amplicons were sequenced to check the presence of potential false-positive results. Basic Local Alignment Search Tool (BLAST; http://www.ncbi.nlm.nih.gov, accessed on 10 December 2020) was used to search homologous sequences and analyses of the sequences obtained corresponding to 305 bp for SaV, 438 bp for EMCV, 570 bp for RVA, and 702 bp AstV, respectively, were found to be identical to each virus. This result showed that the developed mRT-PCR assay is specific.

### 3.4. Assay Reproducibility 

Four mRT-PCR reactions were performed at different times to assess inter-assay reproducibility using the same reaction conditions. A freshly extracted plasmid was used as a template for each PCR reaction. Each of the four mRT-PCR reactions showed a similar result (data not shown), indicating the reproducibility of the assay. Besides inter-assay reproducibility, intra-assay reproducibility was checked by carrying out triplicate mRT-PCR reactions (data not shown). The developed mRT-PCR assay’s intra-assay and inter-assay reproducibility tests showed that it could be used to detect potentially zoonotic swine viruses. 

### 3.5. Detection of Field Samples

A total of 280 porcine fecal samples were tested for SaV, EMCV, RVA and AstV using the developed mRT-PCR assay. One hundred of the 280 tested samples were found to be positive for at least one virus. A positive rate of 24.6%, 5%, 4.3% and 17.5% was observed for SaV, EMCV, RVA and AstV (Table 2). Though only RVA showed a statistically significant difference in prevalence based on age (*p* < 0.05, *X*^2^ = 8), as swine age increases, the remaining three viruses showed a slight decrease in prevalence (Table 2). A statistically significant difference (*p* < 0.05, *X*^2^ = 5.3) in prevalence was found between diarrheic and non-diarrheic pigs. In contrast, the prevalence of the other three viruses did not show any statistically significant difference between diarrheic and non-diarrheic pigs.

With varying degrees of double and triple infection, a mixed infection rate of 6.4% (18/280) was detected. Out of the 100 samples found positive for at least one virus, a coinfection rate of 5% (SaV, RVA and AstV), 4% (SaV, EMCV and AstV), 1% (EMCV, RVA and AstV), 8% (SaV and Astv), 3% (SaV and EMCV), 2% (RVA and AstV) and 1% (RVA and SaV) was observed (Figure 3).

For checking the reliability of the mRT-PCR assay, 130 samples containing 40 samples from diarrheic pigs were selected and further tested using sRT-PCR for SaV, EMCV, RVA and AstV. Except for SaV (Table 3), samples positive for each virus using sRT-PCR test also showed similar positive results when tested by mRT-PCR. Some of the mRT-PCR positive samples were also checked by sequencing. The concordance of sequence results were then checked by subjecting them to NCBI nucleotide blast and no false-positive results were found. 

SaV infection is the most common of the four viruses tested, while RVA infection is the least common. Test results of mRT-PCR showed 98.5% agreement with the test results of sRT-PCR. This result indicated that, like sRT-PCR, the developed mRT-PCR is sensitive enough to detect field samples.

### 3.6. Phylogenetic Analysis 

Phylogenetic trees were constructed using the neighbor-joining method. When constructing the phylogenetic trees, geographic location, host type, and virus genotype were all considered. 

Within the *Caliciviridae* family, based on their complete VP1 nucleotide sequences, sapoviruses could be classified into 15 genogroups, eight of which have been detected in swine (GIII, GV–GXI) [30,31], but only the GI, GII, GIV, and GV genogroups are known to infect humans [30,32,33]. To investigate the genetic diversity of several of the sequences identified in this study, a phylogenetic tree for SaV was constructed based on the partial sequence of the ORF1 region using 5 identified strains (Sapo 1 to Sap 5) and 32 reference strains. The SaV strains identified in this study named (Sapo 1, Sapo 2, Sapo 3 and Sapo 5) clustered within the GIII genogroup and were found to be closely related to other GIII SaV strains detected in China (MW285642) and Mexico (MH490911). Whereas Sapo 4, which is also classified as a GIII genogroup member, formed a separate clade (Figure 4A).

EMCV is a *Picornaviridae* virus that infects many animals, including pigs, mice, primates, and humans [34]. The EMCV strains isolated from various animals, for example, pigs and rats, demonstrated a high degree of homology [35]. Thus, to investigate the genetic variability of the detected EMCV strains, a phylogenetic tree for EMCV was constructed based on the partial sequence of the 3D region using three identified strains (EMCV 1 to EMCV 3) and 22 reference strains from GenBank. According to the partial sequence of the 3D region, EMCV 3 detected in this study clustered with other EMCV strains from Japan (LC508268), whereas EMCV 1 and EMCV 2 clustered separately and more closely related to strains from China (Figure 4B).

A phylogenetic tree for RVA was constructed based on the partial sequence of the VP6 region using three identified strains (R1 to R3) and 30 reference strains from GenBank. The RVA variants identified in this study clustered into two distinct clades, with R1 and R2 belonging to one clade and R3 to another clade. However, all RVA variants identified in this study are more closely related to RVA previously described from China (MT874988, FJ617209 and KT82077), implying that they may have originated from the common ancestor. R1 and R2 clustered with MT874988 and FJ617209, previously described from China, whereas R3 clustered with KT820771, another Chinese RVA strain (Figure 4C). 

Based on the partial sequence of the ORF1ab region, a phylogenetic tree for AstV was constructed using three identified strains (Ast 1 to Ast 7) and 25 reference strains from GenBank. All AstV variants detected during this study clustered under AstV4. Furthermore, Ast 1, 2, 3, 5 and Ast 7 clustered closely with AstV from China (MK460231) and Kenya (MT451918), whereas Ast 4 and Ast 6 clustered independently and close to strain from the USA (JX556692) (Figure 4D).

## 4. Discussion

Despite the presence of a plethora of different types of diagnostic assays, nucleic acid-based assays offer an added advantage of high specificity and sensitivity with the possibility of a lower detection limit [36]. Recently, mRT-PCR is being used as one of the most important diagnostic methods for rapidly and simultaneously detecting viral pathogens. Additionally, hence, different mRT-PCR assays for simultaneously detecting nine [37], six [25,38], five [39], four [23,24,26,40], three [41] and two [42,43] swine enteric viruses have been developed. Due to the asymptomatic nature of SaV, EMCV, RVA, and AstV infections, these viruses have received insufficient attention, and no single diagnostic assay capable of simultaneously detecting these viruses from porcine fecal samples has been developed.

A mRT-PCR assay capable of simultaneously detecting SaV, EMCV, RVA, and AstV RNA from porcine fecal samples was developed in this study. Our assay and other previously developed assays [23,26] for other viruses suggest that sRT-PCR may be more specific than mRT-PCR, but the improvement in turnaround time and cost-effectiveness would compensate for this minor reduction in sensitivity. Yet, compared to sRT-PCR, mRT-PCR is more economical and rapid. The sensitivity of the developed mRT-PCR assay using plasmids containing the specific viral target fragments was 2.5 × 10^3^ copies for each template. Similarly, Zhao et al. [23] reported a similar mRT-PCR assay sensitivity of 2.17 × 10^3^, 2.1 × 10^3^, 1.74 × 10^4^ and 1.26 × 10^4^ for porcine epidemic diarrhea virus, transmissible gastroenteritis virus, RVA, and porcine circovirus 2, respectively. Thus, the currently developed mRT-PCR assay has similar sensitivity with Hu et al. [40] and Liu et al. [26], a higher sensitivity compared to Day et al. [44] and Cagirgan and Yazici [45], and lower sensitivity compared to Liu et al. [39]. Further, 130 fecal samples were tested for checking the difference in sensitivity of sRT-PCR and mRT-PCR assays. Test results of mRT-PCR showed an overall concordance rate of 98.5% (128/130) (Table 3) to the test results of sRT-PCR. Besides sensitivity, test results of specificity and reproducibility of this assay indicate that similar to the sRT-PCR, the developed mRT-PCR could be employed to detect SaV, EMCV, RVA, and AstV from porcine fecal samples. Similarly, previous reports [25,37] and others suggested that mRT-PCR could be a highly sensitive and specific assay that could be used for the rapid detection of swine viruses from fecal specimens.

To further confirm the validity of the developed mRT-PCR assay, 280 porcine fecal samples were tested and the positive rates of SaV, EMCV, RVA and AstV was found to be 24.6% (69/280), 5% (14/280), 4.3% (12/280), and 17.5% (49/280), respectively. This shows that, compared to the other two viruses, SaV has the highest positive rate and RVA the lowest positive rate. Similarly, SaV, EMCV, RVA, and AstV have been detected in diarrheic and non-diarrheic swine feces [5,11,12,13,15], highlighting the importance of detecting these viruses in swine farms. In swine, coinfection of enteric viruses is common. In this study, coinfection rate of 5% (SaV, RVA and AstV), 4% (SaV, EMCV and AstV),1% (EMCV, RVA and AstV), 8% (SaV and Astv), 3% (SaV and EMCV), 2% (RVA and AstV) and 1% (RVA and SaV) has been observed. Similar to the current study, coinfections of pigs with different enteric viruses have been reported from China [25,46], the United States [47] and Belgium [48].

A phylogenetic analysis was also made to explore further the epidemiologic characteristics of the detected viruses. The phylogenetic analysis revealed that the SaV detected in this study belongs to the GIII genogroup. Similar to this study, previous research [49,50] indicated that from the eight SaV genogroups detected in swine [30,31], the GIII genogroup is the most prevalent in China. Phylogenetic analysis of the partial sequence of the 3D region of EMCV, similar to previous studies from China [51,52], revealed that the EMCV variants detected in this study clustered into two groups. Yuan, Song, Zhang, Zhang and Sun [52] assigned the EMCV-30 strain (DQ288856) from the USA and the Korean strains K3 (EU780148) and K11 (EU780149) to the EMCV group one lineage while strains PV2 (X87335) and D variant (M37588) to group two lineage. Based on Yuan, Song, Zhang, Zhang and Sun [52], three of the EMCV strains identified in this study may belong to group one lineage, regardless of their clustering pattern. Based on the VP7 gene sequence, Group A rotaviruses are classified into 36 G genotypes, of which 12 G-genotypes (G1–G6, G8–G12, and G26) have been identified in porcine [18,53]. All three RVA strains detected in this study were found closely grouped with strains NJ2012 (MT874988.1) and TA-3-1 (KT820771.1), both of which are G9 genotypes, implying that the RVA strains detected in this study could be G9 genotype. According to phylogenetic analysis, the AstV variants detected in this study belong to AstV4. Similarly, previous reports from the United States [54], Thailand [55] and Slovakia [8] indicated that AstV4 is the most prevalent type of AstV. In contrast, a higher prevalence of AstV2 from China [56] and AstV5 from China’s Hunan province [57] have also been observed, highlighting the importance of further AstV epidemiology studies. These findings lend credence to the notion that coinfection of these viruses is common in swine. Through recombination and mutations, the coinfection of viruses accelerates the evolution of coinfected viruses [58,59], allowing more virulent virus strains to emerge. As a result, the developed mRT-PCR assay could play a critical role in controlling and preventing SaV, EMCV, RVA, and AstV through early and accurate detection.

## 5. Conclusions

The assay sensitivity, specificity, and repeatability results demonstrated that the developed mRT-PCR could be employed to detect SaV, EMCV, RVA, and AstV simultaneously and rapidly in swine fecal specimens. Potential zoonotic swine viruses such as SaV, EMCV, RVA, and AstV are prevalent in Heilongjiang province, China. These viruses frequently coinfect pigs, and detecting them quickly and cost-effectively may necessitate specialized diagnostic techniques such as mRT-PCR. The current mRT-PCR may assist in lowering the cost and time associated with sample processing and testing during large-scale field sample screening for SaV, EMCV, RVA, and AstV. As a result, this assay may help control and prevent potential zoonotic swine viruses by detecting them early and accurately.

## Figures and Tables

**Figure 1 vetsci-09-00176-f001:**
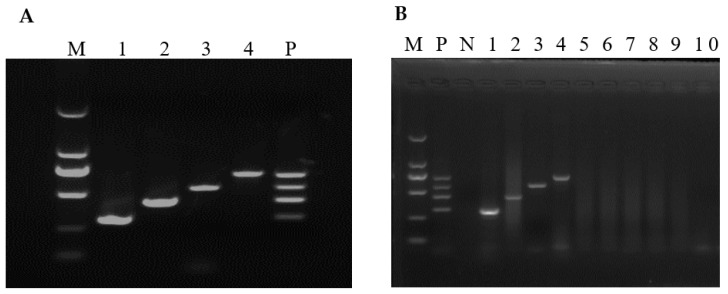
mRT-PCR optimization and assay specificity. M: 2000 bp marker, 1: SaV, 2: EMCV, 3: RVA and 4: AstV, P: standard plasmid template mix containing SaV, EMCV, RVA and AstV fragments. (**A**) Optimization of mRT-PCR using plasmids containing SaV, EMCV, RVA and AstV fragments. (**B**) Assay specificity. *N*: negative control, 1: SaV, 2: EMCV, 3: RVA, 4: AstV and 5–10: PEDV, SVV, TGEV, PRRSV, PDCoV and PSV, respectively.

**Figure 2 vetsci-09-00176-f002:**
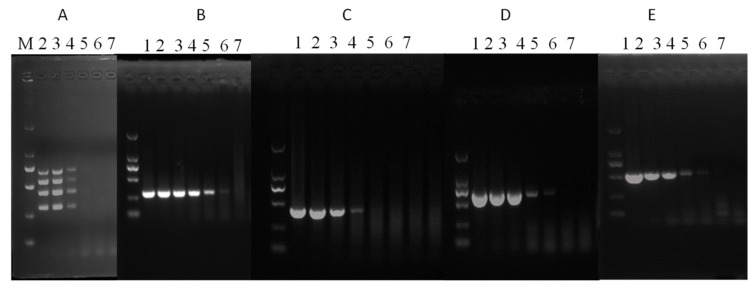
mRT-PCR assay sensitivity. The sensitivity of mRT-PCR and sRT-PCR assays was checked using 10-fold serially diluted plasmids containing SaV, EMCV, RVA and AstV fragments as a template. M: DL2000 DNA Marker, 1: 2.5 × 10^6^ copies, 2: 2.5 × 10^5^ copies, 3: 2.5 × 10^4^ copies, 4: 2.5 × 10^3^ copies, 5: 2.5 × 10^2^ copies, 6: 2.5 × 10^1^ copies and 7: 2.5 × 10^0^ copies. (**A**) mRT-PCR, (**B**) SaV, (**C**) EMCV, (**D**) RVA, and (**E**) AstV.

**Figure 3 vetsci-09-00176-f003:**
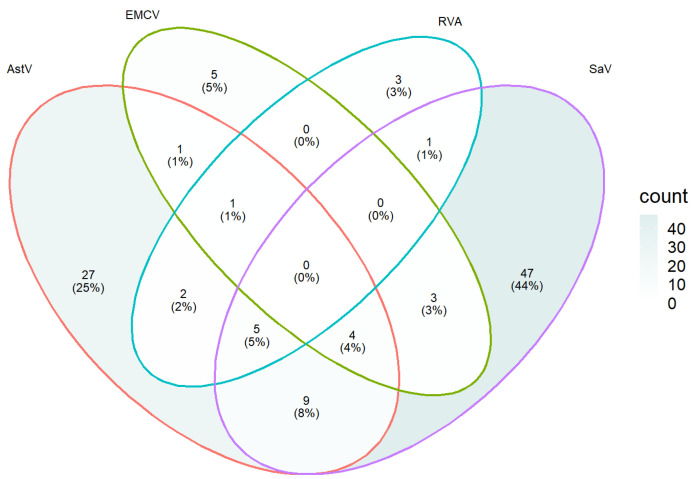
The number of single and mixed infections of Ast, EMCV, RVA and SaV. The coinfection rate of each virus was calculated using 100 samples that tested positive for at least one virus.

**Figure 4 vetsci-09-00176-f004:**
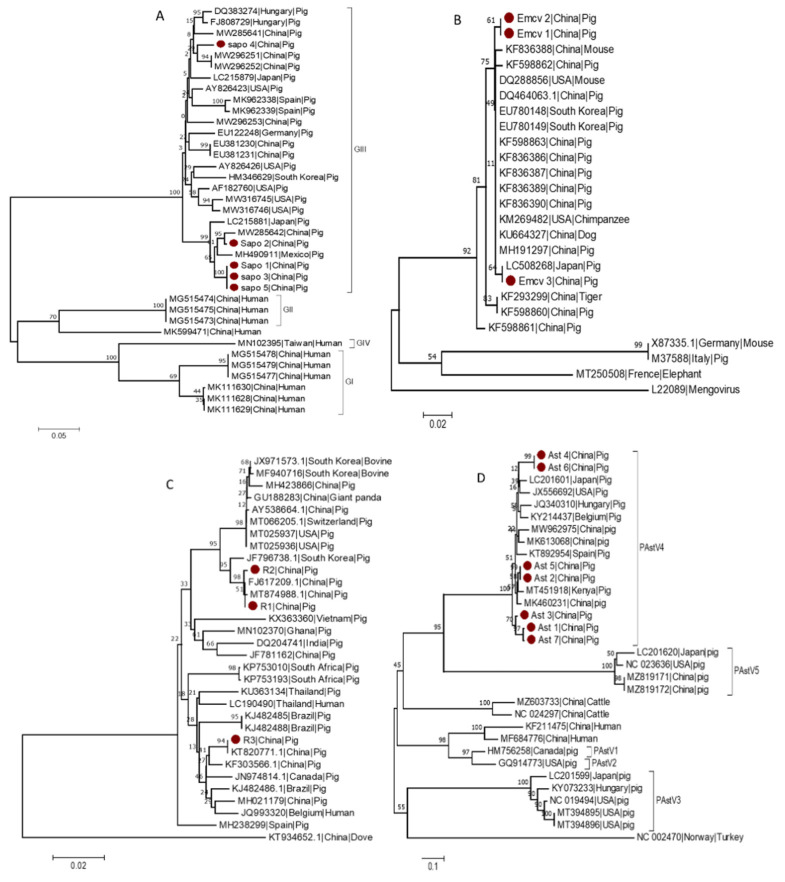
Phylogenetic analysis of SaV, EMCV, RVA and AstV. The strains identified in this study (marked with red dots) and reference strains from GenBank were used to construct phylogenetic trees. Scales indicate units of the number of base substitutions per site. (**A**) Phylogenetic tree of SaV. The tree was built using 224 positions from the ORF1 region of SaV. (**B**) Phylogenetic tree of EMCV. The tree was constructed using 211 positions from the 3D region of the EMCV. Mengovirus was used as an outgroup. (**C**) Phylogenetic tree of RVA. The tree was built based on 545 positions of the VP6 region. Rotavirus A from dove (avian) was used as an outgroup. (**D**) Phylogenetic tree of AstV. The tree was built based on 379 positions of the ORF1ab region. As an outgroup, Turkey (avian) Astrovirus is included.

**Table 1 vetsci-09-00176-t001:** List of primers used to detect SaV, EMCV, RVA and AstV by the mRT-PCR.

Primer	Sequence	Target Gene	Accession No.	Target Region (bp)	Amplicon Size (bp)
SaV-F	AGCCAGAAGTGTTCGTGATGG	ORF1	MK965898.1	5124–5429	306
SaV-R	GGACARGTGRAGYGTGTARGG				
EMCV-F	CCGTCAAGTCTTCCAACCAG	3D	MH191297.1	6288–6725	438
EMCV-R	GCGGCTTGAACCTTCTCTATC				
RVA-F	GCAAACGAAGTCTTCGACATGG	VP6	MH308723.1	6–575	570
RVA-R	GGCGTTAATCCACATAGTYCCCA				
AstV -F	TTGTGGAGCTTGACTGGACC	ORF1ab	MK613068.1	3341–4042	702
AstV-R	CTGTGAGTCTTGCAGGCAGA				

**Table 2 vetsci-09-00176-t002:** Prevalence of AstV, EMCV, RVA and SaV.

Age Group	Health Status	Number of Samples (*n*)	% (Number of Positive/*n*)			
			SaV	EMCV	PRVA	AstV
Piglet	Diarrheic	31	71 (22/31)	19.4 (6/31)	19.4 (6/31)	41.9 (13/31)
	Non-Diarrheic	41	7.3 (3/41)	2.4 (1/41)	2.4 (1/41)	17.1 (7/41)
	Sub-total	72	34.7 (25/72)	9.7 (7/72)	9.7 (7/72)	27.8 (20/72)
Weaner	Diarrheic	12	75 (9/12)	0.0	16.7 (2/12)	50 (6/12)
	Non-Diarrheic	30	10 (3/30)	13.3 (4/30)	3.3 (1/30)	16.7 (5/30)
	Sub-total	42	28.6 (12/42)	9.5 (4/42)	7.1 (3/42)	26.2 (11/42)
Fattening pig	Diarrheic	20	80 (16/20)	15 (3/20)	5 (1/20)	30 (6/20)
	Non-Diarrheic	42	2.4 (1/42)	0	0	7.1 (3/42)
	Sub-total	62	27.4 (17/62)	4.8 (3/62)	1.6 (1/62)	14.5 (9/62)
Sow	Diarrheic	19	78.9 (15/19)	0	5.3 (1/19)	26.3 (5/19)
	Non-Diarrheic	85	0	0	0	4.7 (4/85)
	Sub-total	104	14.4 (15/104)	0.0	1 (1/104)	8.7 (9/104)
		280	24.6 (69/280)	5 (14/280)	4.3 (12/280)	17.5 (49/280)

**Table 3 vetsci-09-00176-t003:** Detection of four viruses in 130 field samples by sRT-PCR and mRT-PCR.

Assay	Number of Positive Samples			
	SaV	EMCV	RVA	AstV
sRT-PCR	29	8	2	13
mRT-PCR	27 *	8	2	13

* The results were similar except for two samples that were positive for SaV by sRT-PCR, but negative by mRT-PCR.

## Data Availability

The data presented in this study can be found in the manuscript.

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
