# Peer review of "Development of a Multiplex RT-PCR Assay for Simultaneous Detection of Four Potential Zoonotic Swine RNA Viruses"

_vetsci, 2022, doi:10.3390/vetsci9040176_

Round 1
Reviewer 1 Report
The manuscript by Weird et al., describes the development of a multiplex RT-PCR for detection of 4 RNA viruses of porcine origin: Sapovirus, astrovirus, encephalomyocarditis virus, and rotavirus A that are suspected to have zoonotic potential. The study is designed and executed clearly with results presented explicitly.
Specific comments:
1) Why was the sensitivity of the mRT-PCR assay checked with fecal samples spiked with the template DNA? Since fecal material might have inhibitors of enzymes, the efficiency and sensitivity of the assay are generally lower compared to the testing with the clean template. Please consider performing this experiment.
2) No mention of checking of primer for auto and heterodimers is mentioned in the methods.
3) Was the detection limit of the multiplex assay tested for different combinations of the target viruses (duplex and triplex)?
Reviewer 2 Report
This is a well-written research paper on the development of multiplex-RT PCR for the identification of neglected swine viruses including porcine Sapovirus (SaV), porcine encephalomyocarditis virus (EMCV), porcine rotavirus A (RVA) and porcine astroviruses (AstV). The viruses cause rather sub-clinical infection in infected pigs, therefore, their diagnosis is very difficult. The authors developed a multplex-PCR for the simultaneous detection of SaV, EMCV, RVA, and AstV from swine fecal samples. The positive rates for SaV, EMCV, RVA, and AstV 24 were found to be 24.6% (69/280), 5% (14/280), 4.3% (12/280), and 17.5% (49/280), respectively. The paper presents new information and a protocol for reliable identification of porcine viruses.
